# ⚡FLASHPLANNER: ACCELERATING DIFFUSION-BASED PLANNER FOR AUTONOMOUS DRIVING VIA GLOBALLY CONSISTENT VELOCITY FIELD AND REDUNDANCY REDUCTION

## ABSTRACT

Standard diffusion and flow matching approaches have recently been explored as imitation-based planners for autonomous driving due to their ability to produce multi-modal trajectories with high fidelity. However, these methods still suffer from limitations, e.g., low efficiency and reliance on post-processing. These issues are alleviated through practices from conventional imitation-based methods, but the principles of well-designed diffusion-based planners are still underexplored. In this paper, we propose *FlashPlanner*, a flow-matching-based planner for online planning of autonomous driving. *FlashPlanner* introduces a novel globally consistent velocity field as the training objective, which frames flow matching to model instantaneous dynamics in a consistent velocity field. This training objective manages to unleash the potential of diffusion-based planners and enables stable one-step generation of high-quality trajectories in closed-loop planning. Moreover, we systematically analyze the existing design choices of diffusion-based methods and prune inherent redundancy, which further accelerates the diffusion-based planning. *FlashPlanner* achieves **state-of-the-art** performance on the closed-loop nuPlan benchmark and delivers **12×** faster inference (**166FPS**) compared to the existing best baseline (13FPS). We will open-source our project.

## 1 INTRODUCTION

Learning-based planning (Jaeger et al., 2025; Hu et al., 2023) has emerged as a promising approach for autonomous driving (AD), offering scalable and flexible advantages over rule-based methods. In particular, imitation-based (Cheng et al., 2024b;a) planners achieve remarkable success based on the availability of large-scale data today. However, conventional imitation-based methods struggle with challenges such as distribution shift and unimodality. The following works have attempted to address these challenges by introducing many techniques, yet they still rely on rule-based post-processing to refine or select good trajectories (Cheng et al., 2024a). This suggests that these methods still fall short of effectively modeling the expert policy in the dataset.

Recently, diffusion-based planners have been popular for planning tasks in robotics (Black et al.; Chi et al., 2023). The multi-modality and high-quality outcomes of diffusion-based planners advance the performance of the imitation-based paradigm. However, when applied in autonomous driving, several methods encounter obstacles like low efficiency, mode collapse, and trajectory divergence (Yang et al., 2024; Liao et al., 2025; Xing et al., 2025). These obstacles are alleviated through practices from conventional imitation-based methods, such as introducing prior anchors, imposing auxiliary losses, and jointly modeling the ego and other agents (Liao et al., 2025; Zheng et al., 2025). These designs introduce additional computation, while the core challenges of imitation learning often persist. On the other hand, diffusion models face an enduring challenge: the outstanding output quality and significant computational cost of the iterative denoise/flow steps (Frans et al., 2024). Recent methods, especially in the image generation task, preserve output quality while reducing sampling steps by introducing more appropriate training objectives, such as consistent endpoint or averaged velocity prediction (Song et al., 2023; Geng et al., 2025; Stability AI, 2025). Nevertheless, the suit-

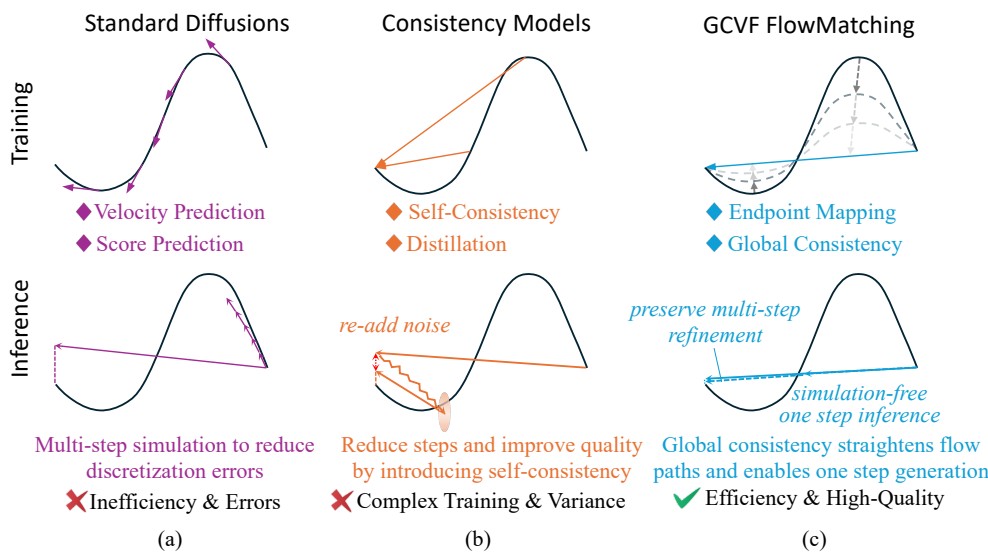

Figure 1: **Different training objectives of diffusion models**. (a) Standard diffusion models learn the score or local instantaneous velocity, so reducing denoising/flow steps for acceleration will yield large discretization errors. (b) Consistency models enable few-step generation through self-consistency or distillation, but often introduce complex training procedures, extra variance, or accumulated errors. (c) GCVF takes the global average velocity as the consistent velocity field and enforces alignment of the instantaneous velocity with this velocity field along the marginal path. The global consistency of GCVF straightens the flow paths, enabling one-step generation while preserving monotonic performance improvement with increased flow steps.

able learning objectives for planning tasks remain underexplored, falling short of fully harnessing the potential of diffusion-based planners. We refer readers to Appendix A.1 for related works A.1.

To address these limitations, we propose FlashPlanner, a lightweight and efficient flow-matching-based planner for real-time planning of autonomous driving. FlashPlanner introduces a *Globally Consistent Velocity Field* (GCVF) as the training objective, which frames flow matching to model instantaneous dynamics in a consistent velocity field ( Figure 1,c). This objective imposes a global consistency constraint across different intermediate flow steps, which straightens the flow paths during training. Compared with score prediction or velocity prediction in standard diffusion models ( Figure 1,a), consistency constraints of GCVF substantially avoid discretization errors when using few flow steps. Moreover, different from finite-horizon training used in consistency models (Song et al., 2023; Geng et al., 2025), GCVF avoids a complex training procedure and diminishes the extra variance and accumulated errors introduced by re-adding noise during generation ( Figure 1,b). GCVF enables the one-step generation of high-quality trajectories for autonomous driving in diverse real-world scenarios, while supporting refinement with more inference steps. (see Figure 6 and Figure 9). We empirically find that **our training objective unleashes the potential of diffusion-based planners, while a simple MLP-based structure can achieve state-of-the-art (SOTA) performance** (see Table 1 and Table 3). We further revisit the prevalent designs of diffusion-based methods, which are inherited from conventional learning-based planners, and reveal the essential components for trajectory generation. Guided by this systematic analysis, we remove redundant designs and devise a lightweight architecture tailored to ego trajectory generation. FlashPlanner achieves superior closed-loop performance among learning-based baselines on the nuPlan, a large-scale real-world autonomous planning benchmark (Caesar et al., 2021). In reactive mode, it exceeds Diffusion Planner (Zheng et al., 2025) by **5.56%** on Test-14 hard split and by **2.04%** on Val-14 hard split. FlashPlanner also shows outstanding training and inference efficiency, delivering **166 FPS** inference speed and substantially reducing training time to **2.5 h**. In summary, our contributions are as follow:

- We propose a flow-based autonomous driving planner, FlashPlanner, which introduces a consistent velocity prediction as the training objective. FlashPlanner enables one-step generation of high-quality trajectories for autonomous driving, while supporting refinement with more inference steps.

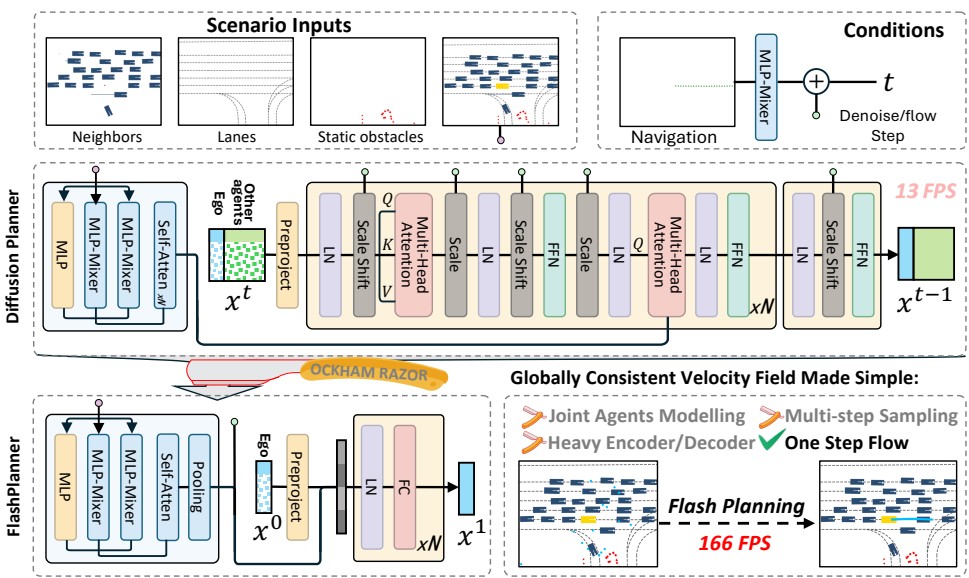

Figure 2: **Model architecture of FlashPlanner and comparison with Diffusion Planner.** The globally consistent velocity field unleashes the potential of diffusion-based planners. It enables pruning of redundant components and yields super-efficient, high-quality trajectory generation.

- We systematically analyze the design choices in existing diffusion-based planners and prune redundant components, e.g., joint modeling agents and overly complex encoder/decoder, leading to a light and robust architecture for diffusion-based online real-time planning.
- FlashPlanner achieves SOTA performance on the closed-loop nuPlan benchmark and delivers a speedup of more than 12× compared with the best existing approach (Zheng et al., 2025).

## 2 PRELIMINARIES

Flow Matching (FM) is a class of generative transport models that learns to transform samples from a known initial distribution $\pi_0$ to a target distribution $\pi_1$, by modeling the time-dependent velocity field of probability flows. Different from standard diffusion models that usually rely on complex noise schedules, FM is simulation-free and is easier for training (Xing et al., 2025; Zhang et al., 2025; Park et al., 2025). For samples $x_0 \sim \pi_0$ and $x_1 \sim \pi_1$, the flow path is characterized by intermediate states $x_t$ where $t \in [0, 1]$, with the instantaneous velocity defined as

$$v_t = \frac{dx_t}{dt}. \tag{1}$$

A neural network $v_\theta$ is trained to approximate this velocity field by minimizing the loss function:

$$L(\theta) = \mathbb{E}_{t,x_0,x_1}\|v_\theta(x_t, t) - v_t\|^2. \tag{2}$$

The learned velocity predictor $v_\theta$ enables sample generation by solving the ordinary differential equation (ODE) $\frac{dx_t}{dt} = v_\theta(x_t, t)$ starting from $x_0 \sim \pi_0$, yielding the solution:

$$x_1 = x_0 + \int_0^1 v_\theta(x_t, t)dt. \tag{3}$$

A practical instantiation is Rectified Flow (Liu et al., 2022), which defines the path via optimal transport displacement, where $\pi_0$ is typically the standard normal distribution $\mathcal{N}(0, I)$.

$$x_t = (1 - t)x_0 + tx_1, \tag{4}$$

## 3 METHOD

In this section, we present FlashPlanner, a flow-based planner that enables one-step generation. We begin by introducing our proposed training objective, *Globally Consistent Velocity Field* (GCVF),

for flow matching. We then reformulate the GCVF-based planning task for autonomous driving. Building on this foundation, we finally introduce the lightweight and robust architecture of Flash-Planner, which is obtained through a detailed revisiting of design choices in existing diffusion-based planners and removing redundant components.

### 3.1 GLOBALLY CONSISTENT VELOCITY FIELD FOR MATCHING

Straight ODE flow paths have been demonstrated to be effective for one-step and few-step generation, since learned straight flow paths substantially reduce discretization error (Park et al., 2024; Liu et al., 2022). While Rectified Flow constructs straight conditional paths as in Equation (4), the learned marginal velocity field $v_\theta$ in Equation (2) typically induces curved marginal paths, resulting in a performance degradation when reducing flow steps (Liu et al., 2022; Geng et al., 2025). To tackle this issue, Rectified Flow (Liu et al., 2022) uses iterative retraining to straighten the flow paths, but it requires additional training and offers limited improvement. Our method, *Globally Consistent Velocity Field* (GCVF), addresses this problem by enforcing alignment between the instantaneous and average velocity along the marginal path.

The instantaneous velocity is defined in Equation (1). We introduce the average velocity over an interval $[t, 1]$:

$$u_t = \frac{x_1 - x_t}{1 - t},$$
(5)

which is the displacement from any intermediate state $x_t$ to the endpoint $x_1$ divided by the remaining time interval. We parameterize a neural network as a data predictor $f_\theta(x_t, t)$ that directly approximates the clean endpoint $x_1$. The training objective is

$$\mathcal{L}(\theta) = \mathbb{E}[||v_t - u_t(\theta)||^2] = \mathbb{E}\left[\left\|\frac{dx_t}{dt} - \frac{f_\theta(x_t, t) - x_t}{1 - t}\right\|^2\right],$$
(6)

which explicitly enforces equality between instantaneous and average velocities, producing straight flow paths. Given the conditional flow in Equation (4), it reduces to the endpoint regression:

$$\mathcal{L}(\theta) = \mathbb{E}\left[\|f_\theta(x_t, t) - x_1\|^2\right].$$
(7)

Thus, the training requires only supervised prediction of $x_1$ from intermediate states, for stable optimization. E.g., the ODE solution in Equation (3) can be reformulated regarding the data predictor:

$$x_1 = x_0 + \int_0^1 (f_\theta(x_t, t) - x_0)dt.$$
(8)

GCVF presents a simple yet effective approach that enables few-step generation through endpoint approximation. Our empirical evaluation demonstrates that, when applied to trajectory generation of autonomous driving, GCVF achieves higher planning scores than standard Flow Matching (Lipman et al., 2022) (Equation (2)) with exceptional efficiency in both training and inference.

**Relation to MeanFlow:** The concept of average velocity is also introduced in MeanFlow (Geng et al., 2025), where it is defined between two time steps $t$ and $r$ as:

$$u(z_t, r, t) = \frac{1}{t - r} \int_r^t v(z_\tau, \tau)d\tau.$$
(9)

MeanFlow calculates the displacement by integrating instantaneous velocities, whereas our definition in Equation (5) computes it as the difference between two endpoints. Based on this distinction, MeanFlow employs neural networks to directly approximate the average velocity, whereas GCVF uses networks to predict the endpoint, indirectly deriving the average velocity. Furthermore, MeanFlow establishes its training objective via the functional relationship between instantaneous and average velocities naturally derived from Equation (9), without additional assumptions. Notably, this relationship holds universally for both curved and straight flow paths. The one-step generation capability of MeanFlow relies on an accurate approximation of average velocities between start and end points. In contrast, GCVF imposes the constraint that instantaneous velocities along the flow path equal average velocities, thereby achieving one-step generation through straight flow paths. The performance comparison between MeanFlow and GCVF for autonomous driving trajectory generation is demonstrated in Figure 8 and Table 6.

## 3.2 TASK FORMULATION

Autonomous driving motion planning takes processed environmental context as input and produces safe, feasible, and optimal future trajectories for the ego-vehicle to follow (Cheng et al., 2024b). Leveraging the strong expressive power of generative models, we propose FlashPlanner for this task, which generates multi-modal, high-quality trajectories in real-time.

The trajectory is represented as a sequence of states $\mathbf{x} = \{(x^\tau, y^\tau, \theta^\tau)\}_{\tau=1}^T$, where $T$ denotes the planning horizon, $\theta^\tau$ is the heading angle, and $(x^\tau, y^\tau)$ indicates the location at time $\tau$ in the current ego-vehicle coordinate system. We adopt the convention that subscripts index the continuous flow-matching time $t \in [0, 1]$, while superscripts index the discrete trajectory timesteps $\tau \in \{1, \ldots, T\}$. FlashPlanner directly predicts the clean trajectory $\mathbf{x}_1$ from a partially noised sample $\mathbf{x}_t$. Drawing from Equation (7), the training objective is:

$$\mathcal{L}(\theta) = \mathbb{E}\left[||f_\theta(\mathbf{x}_t, t, \mathbf{C}) - \mathbf{x}_1||^2\right], \tag{10}$$

where $\mathbf{x}_1$ represents the expert trajectories used for supervision, and $\mathbf{C}$ denotes the environmental context about current vehicle states, historical data, lane information, and navigation information.

During inference, FlashPlanner generates multi-modal trajectories by transporting samples from an initial noise distribution to the trajectory distribution conditioned on the environmental context. We solve the corresponding flow ODE using the Euler method, yielding the final trajectory as:

$$\mathbf{x}_1 = \mathbf{x}_0 + \sum_{i=1}^n \frac{1}{n} \left[f_\theta(\mathbf{x}_{t_i}, t_i, \mathbf{C}) - \mathbf{x}_0\right], \tag{11}$$

where $n$ is the total number of inference steps, and $t_i \in [0, 1]$ is the flow time at the i-th step. In practice, $n = 1$ is sufficient to produce high-quality trajectories. By sampling different initial noise $\mathbf{x}_0$, FlashPlanner can generate diverse future trajectories.

## 3.3 REDUNDANCY REDUCTION STRATEGIES

Existing diffusion-based planners often incorporate design choices inherited from conventional imitation-based methods, e.g., heavy transformer-based networks, joint modeling of both prediction and planning tasks, and introducing auxiliary loss (Yang et al., 2024; Zheng et al., 2025). For example, Diffusion Planner (Zheng et al., 2025), one of the best diffusion-based AD planners, jointly models the motion prediction and closed-loop planning tasks as a trajectory generation task inspired by prior works (Ngiam et al., 2021; Hu et al., 2023). However, this design involves additional computation and convergence difficulties, while its effectiveness has not been fully validated. In other words, many of these designs are likely redundant, and the existing diffusion-based planners keep them because the potential of generative modeling is not fully realized. Given that our

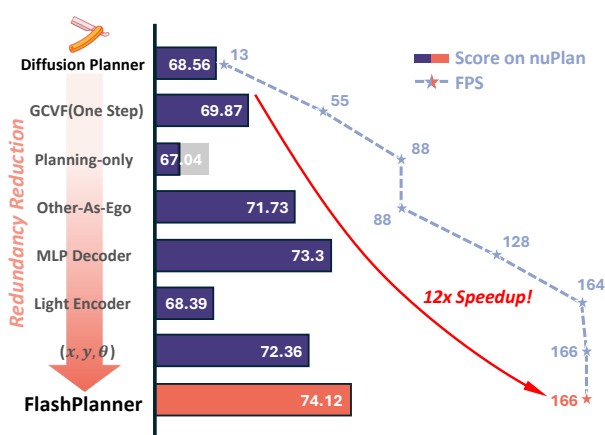

Figure 3: **Redundancy reduction roadmap.** Step-by-step evolution from Diffusion Planner (baseline) to FlashPlanner (ours), illustrating the cumulative impact of our design choices on nuPlan Test14-hard (Reactive mode) performance and inference throughput. Detailed description of each variant is provided in Appendix A.2. All experiments were conducted on the same device.

GCVF offers stronger approximation capacity and robustness, we boldly remove a series of redundant designs and empirically verify their redundancy in diffusion-based planners (see Figure 3 and Table 7). As a result, we obtain a lightweight yet powerful diffusion-based planner with real-time performance. Accordingly, we first describe the redundant structures we removed, and then present our lightweight architecture.

**Planning-only objective:** Trajectories of neighboring vehicles are useful for the planning task, but joint modeling of traffic prediction and ego planning in the same network introduces unnecessary computational complexity and requires additional modules for multi-agent interaction representation. Inspired by Chen & Krähenbühl (2022), we focus exclusively on ego planning while leveraging other vehicles' trajectories as additional ego samples for training. Specifically, we densify supervision by treating other agents as ego vehicles via coordinate transformation. This approach utilizes other agents' behaviors to create more diverse, challenging driving scenarios without extra data collection, thereby enhancing sample efficiency and generalization capability.

**MLP vs. Transformer:** Prior diffusion-based planners typically employ heavy transformer-based decoders with multi-layer self-attention to capture interactions between the ego trajectory and the environment (Zheng et al., 2025; Yang et al., 2024). However, recent advances in related domains (Chen et al., 2025) have demonstrated that appropriate objective design and architecture enable lightweight networks to deliver robust performance. Inspired by that, we developed an extremely lightweight MLP-based decoder for FlashPlanner. It replaces the complex transformer structure and condition mechanism with several fully-connected layers and layer-norm layers. We empirically find that a compact MLP-based decoder is sufficient for high-quality trajectory generation while substantially reducing computational overhead.

**Other Reduction Strategies**. We further investigate more components, such as data augmentation, the depth of the encoder, and the structure of the predicted vector. Figure 3 provides an overview of the design choices introduced in FlashPlanner, which reveal the components for trajectory generation. Due to space limit, detailed analysis of each component is presented in Appendix A.5.

## 3.4 LIGHTWEIGHT MODEL ARCHITECTURE

FlashPlanner presents a lightweight MLP-centric architecture, featuring a simple yet effective mechanism for trajectory generation. Figure 2 and Figure 4 illustrate an overview of the complete architecture, and a detailed description is provided as follows.

**Encoder.** For a driving scene at the current timestep, the encoder inputs consist of the historical trajectories of $N_d$ dynamic agents over $T_h$ timestamps, the status of $N_s$ static obstacles, and the information of $N_l$ lanes, denoted as $\mathbf{F}_d \in \mathbb{R}^{N_d \times T_h \times D_d}$, $\mathbf{F}_s \in \mathbb{R}^{N_s \times D_s}$, and $\mathbf{F}_l \in \mathbb{R}^{N_l \times P_l \times D_l}$, respectively, where $P_l$ represents the number of waypoints per lane polyline. The feature dimension $D_d$ contains agent coordinates, heading, velocity, size, and category. $D_s$ is analogous but excludes velocity. $D_l$ provides lane details encompassing waypoint coordinates, traffic light status, and speed limits. Following the encoder architecture of Diffusion Planner (Zheng et al., 2025), each modality is initially embedded by modality-specific MLPs and processed with MixerBlocks (Tolstikhin et al., 2021) for feature extraction and temporal-spatial fusion. Subsequently, the encoded representations are concatenated across modalities and passed through a multi-head self-attention block to model cross-modal interactions, producing a comprehensive scene encoding $\mathbf{F} \in \mathbb{R}^{(N_d+N_s+N_l) \times D}$, where $D$ is the hidden dimension. In contrast to the multi-layer stacking used in Diffusion Planner, we adopt a more efficient design featuring only one MixerBlock per modality and a single self-attention layer, achieving significant parameter reduction while maintaining representation capabilities.

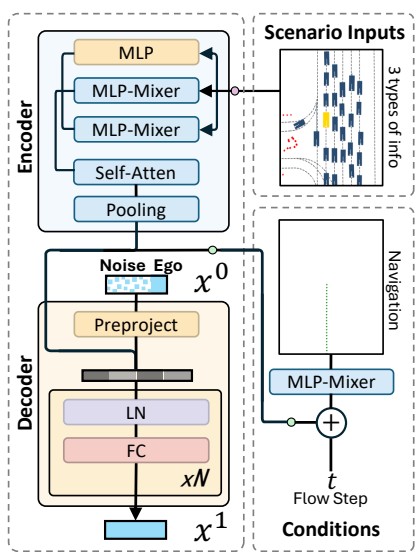

Figure 4: FlashPlanner's Reillustration.

**Decoder.** It integrates scene context, navigation information, noised ego future trajectory, and flow timestep to generate ego future trajectory. Specifically, the scene encoding $\mathbf{F}$ from the encoder is aggregated via mean pooling, yielding a compact scene representation $\mathbf{F}' \in \mathbb{R}^D$. The navigation information, which specifies the intended route and provides critical guidance for motion planning, is denoted as $\mathbf{F}_r \in \mathbb{R}^{N_r \times P_l \times D_l}$ where $N_r$ indicates the number of lanes in the route. We encode $\mathbf{F}_r$ through MLP and MixerBlock to obtain the route embedding $\mathbf{F}'_r \in \mathbb{R}^D$. The flow timestep $t$ is

Table 1: Closed-loop planning results on nuPlan dataset.

| Type | Planner | Val14 | | Test14-hard | |
|---|---|---|---|---|---|
| | | NR | R | NR | R |
| Expert | Log-replay | 93.53 | 80.32 | 85.96 | 68.80 |
| Rule-based & Hybrid | IDM | 75.60 | 77.33 | 56.15 | 62.26 |
| | PDM-Closed | 92.84 | 92.12 | 65.08 | 75.19 |
| | PDM-Hybrid | 92.77 | 92.11 | 65.99 | 76.07 |
| | GameFormer | 79.94 | 79.78 | 68.70 | 67.05 |
| | PLUTO | 92.88 | 76.88 | 80.08 | 76.88 |
| | Diffusion Planner w/ refine. | 94.26 | 92.90 | 78.87 | 82.00 |
| | FlashPlanner w/ refine. (Ours) | 94.97 | 93.12 | 80.83 | 82.71 |
| Learning-based | PDM-Open | 53.53 | 54.24 | 33.51 | 35.83 |
| | UrbanDriver | 68.57 | 64.11 | 50.40 | 49.95 |
| | GameFormer w/o refine. | 13.32 | 8.69 | 7.08 | 6.69 |
| | PlanTF | 84.27 | 76.95 | 69.70 | 61.61 |
| | PLUTO w/o refine. | 88.89 | 78.11 | 70.03 | 59.74 |
| | Diffusion Planner | 89.76 | 82.56 | 75.67 | 68.56 |
| | FlashPlanner (Ours) | 89.19 | 84.60 | 76.46 | 74.12 |

mapped to a fixed-dimensional embedding $\mathbf{T} \in \mathbb{R}^D$ via sinusoidal positional encoding. The noised ego future trajectory $\mathbf{X}_t \in \mathbb{R}^{T_f \times 3}$ is projected to $\mathbf{X}'_t \in \mathbb{R}^D$ using MLP. Finally, the four embeddings $[\mathbf{F}', \mathbf{F}'_r, \mathbf{T}, \mathbf{X}'_t]$ are concatenated and processed through a lightweight MLP head to generate the denoised ego future trajectory, achieving effective multi-modal information fusion efficiently.

## 4 EXPERIMENTS

### 4.1 EXPERIMENT SETUP

**Dataset and Benchmark.** We train and evaluate FlashPlanner on the nuPlan dataset (Caesar et al., 2021), which provides the first publicly accessible, large-scale planning benchmark for autonomous driving with its associated closed-loop simulation framework. For training, we employ a standardized split comprising 1 million frames sampled across all scenario types to ensure broad coverage of driving contexts. For evaluation, we conduct extensive experiments on both the Test14-hard and Val14 benchmarks under non-reactive (NR) and reactive (R) modes. The closed-loop score is adopted as our primary metric, which is a composite measure that aggregates progress, time-to-collision, speed-limit compliance, and comfort into a unified score ranging from 0 to 100, with higher values indicating better planning performance.

**Baseline.** We conduct comparisons between FlashPlanner and existing state-of-the-art models on the nuPlan benchmark. The baselines are categorized into three groups: **Rule-based** methods that rely on manually engineered rules, **Learning-based** methods that employ neural networks to directly output the final planned trajectories, and **Hybrid** methods that incorporate post-processing modules to refine learning-based outputs. Detailed descriptions of each baseline are given in Appendix A.3. The implementation details of FlashPlanner can be found in Section A.4.

### 4.2 MAIN RESULTS

We conduct evaluations of FlashPlanner with state-of-the-art planning methods on the nuPlan benchmark, with the quantitative results presented in Table 1. Among learning-based approaches, Flash-Planner achieves superior overall performance across key metrics, with particularly pronounced gains in the reactive setting. While both FlashPlanner and Diffusion Planner employ generative modeling frameworks and deliver comparable non-reactive results, FlashPlanner exhibits substantial improvements in reactive scenarios, outperforming Diffusion Planner by +2.04 on Val14 and +5.56 on Test14-hard, indicating enhanced robustness to interaction-induced disturbances.

Although hybrid methods such as PDM-Closed and PDM-Hybrid perform competitively on Val14, they exhibit limited generalization on the more challenging Test14-hard benchmark. In contrast, FlashPlanner maintains consistently strong performance across both nominal and interaction-heavy scenarios, even surpassing the expert-level baseline in certain configurations. We attribute these

Table 2: **Efficiency of FlashPlanner vs. Diffusion Planner.** Experiments on the same device.

| Method | Val14 | | Test14-hard | | Inference | Training |
|---|---|---|---|---|---|---|
| | NR | R | NR | R | FPS | Time (h) |
| Diffusion Planner (500 epochs) | 89.76 | 82.56 | 75.67 | 68.56 | 13.7 | 38.3 |
| Diffusion Planner (100 epochs) | 86.87 | 76.30 | 74.62 | 62.25 | 13.7 | 7.7 |
| FlashPlanner (100 epochs) | 89.19 | 84.60 | 76.46 | 74.12 | 166.7 | 2.5 |
| Speedup | - | - | - | - | 12.2× | 15.3× |

gains to the model's ability to generate high-quality trajectories that are further enhanced by refinement, effectively bridging the performance gap between learning-based and hybrid approaches.

Table 2 reports an efficiency comparison between FlashPlanner and Diffusion Planner, the prior best learning-based approach. Additional comparisons with learning-based planners are provided in Table 5. For fairness, Diffusion Planner is evaluated under two training budgets: 500 epochs (original) and 100 epochs, with all other experimental settings unchanged. Inference time denotes the per-frame latency in non-reactive mode (seconds), and training time is measured in hours.

| FlashPlanner (ours) | Diffusion Planner | PLUTO | PlanTF |
|---|---|---|---|

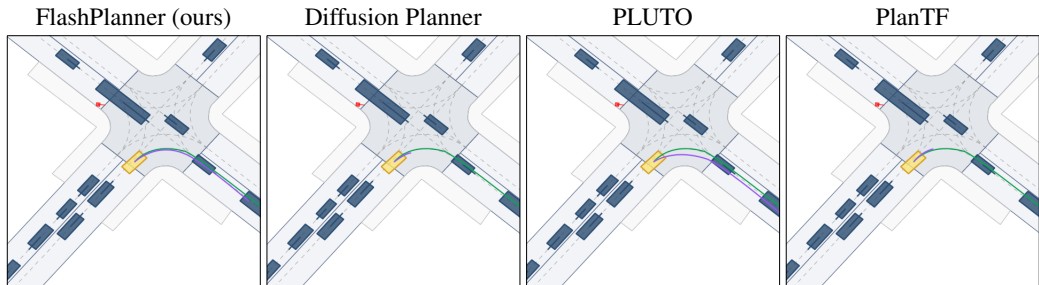

Figure 5: **Future trajectory visualization in closed-loop evaluation**. Trajectory planning results for a challenging narrow-road turning scenario, showing the planned future trajectory and the ground-truth of the ego vehicle.

Benefiting from our architectural design and parameterization choice, FlashPlanner achieves 166 FPS inference speed and completes training in 2.5 hours. Compared to the fully trained Diffusion Planner, FlashPlanner delivers 12× faster inference and 15× faster training while maintaining superior performance across most metrics, with only a marginally lower score on Val14 NR. When both models are trained for 100 epochs, FlashPlanner significantly outperforms Diffusion Planner across all planning metrics. These results show the substantial efficiency of FlashPlanner, making it practical for real-world deployment.

To further show the capabilities of learning-based planners, we visualize planned trajectories of representative baselines in Figure 5. FlashPlanner exhibits high-quality trajectory generation with smooth, kinematically feasible paths. In contrast, Diffusion Planner and PlanTF tend to become nearly stationary given dense traffic ahead, reflecting overly conservative behavior and limited dynamic interaction modeling. PLUTO produces trajectories with abrupt steering maneuvers and rapid heading changes, indicating large curvature and high yaw-rate that compromise comfort and lateral stability. Overall, FlashPlanner achieves a more favorable balance of safety, efficiency, and comfort.

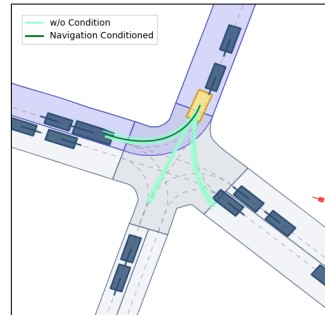

Figure 6: Multi-modal planning behaviors of FlashPlanner.

To evaluate FlashPlanner's multi-modal planning capabilities, we conduct multiple inference runs from the same initial state in an intersection scenario. As shown in Figure 6, without navigation guidance, FlashPlanner generates diverse trajectory distributions corresponding to three distinct maneuvers: left turn, right turn, and straight. When provided with navigation constraints, the planner reliably adheres to the specified route. These results demonstrate that FlashPlanner effectively captures multi-modal driving behaviors and can generate trajectories following conditional guidance.

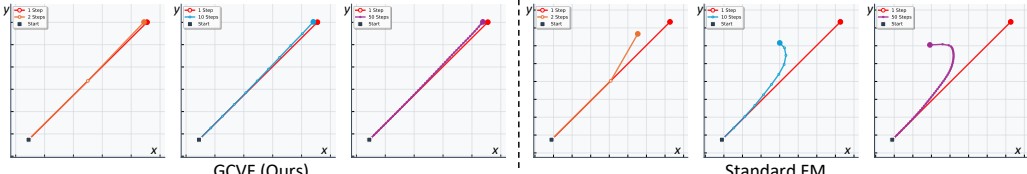

Figure 7: **Flow paths visualization.** Illustration of the generation process for GCVF and Standard FM across different inference steps (1, 2, 10, 50), depicting the evolution from the same initial point sampled from $\mathcal{N}(0, I)$ to the final planned state.

### 4.3 ABLATION STUDY

We demonstrate the effectiveness of two key components, decoder architecture and training objective, as shown in Table 3. Ablation studies for other design choices are given in Appendix A.5.

**Decoder Architecture:** Despite the wide use of Transformer-based decoders in learning-based motion planning, our ablations indicate that such a choice is unnecessary and inefficient in this setting. FlashPlanner employs a lightweight MLP-based decoder that delivers superior efficiency and planning performance compared to Transformer alternatives. Under identical configurations with 100 training epochs, the MLP decoder consistently outperforms its Transformer-based counterpart across both training objectives: 76.46 vs. 74.93 on GCVF and 66.56 vs. 27.83 on Standard FM. It also demonstrates faster convergence: the MLP attains strong performance with 100 epochs, whereas the Transformer requires ex-

Table 3: **Ablation study on decoder architecture and training objective.** Scores are reported on nuPlan Test14-hard (NR); the training objectives of GCVF and Standard FM correspond to Equation (10) and Equation (2), respectively.

| Decoder | Training Objective | Epochs | Steps | Score |
|---|---|---|---|---|
| MLP | GCVF | 100 | 1 | 76.46 |
| | Standard FM | 100 | 1 | 66.56 |
| | | | 10 | 75.57 |
| Transformer | GCVF | 100 | 1 | 74.93 |
| | | 500 | 1 | 76.08 |
| | Standard FM | 100 | 1 | 27.83 |
| | | | 10 | 36.26 |
| | | 500 | 1 | 56.63 |
| | | | 10 | 66.07 |

tended training up to 500 epochs for comparable results. This efficiency advantage is particularly evident on Standard FM, where the MLP at 100 epochs (66.56) exceeds the Transformer even at 500 epochs (56.63), underscoring the superior learning efficiency of our simplified architecture for autonomous driving planning.

**Training Objective:** GCVF demonstrates superior efficiency in both training and inference, achieving optimal planning performance with minimal computational cost. In contrast, Standard FM requires both intensive training and multi-step inference to reach comparable performance. At inference, Standard FM attains an acceptable score with 10 flow steps, yet still underperforms GCVF's single-step results. Training Standard FM is also more challenging, especially with Transformer-based decoders: extending training from 100 to 500 epochs increases the score from 28.79 to 57.59, indicating that substantially longer training is needed to approach convergence.

As illustrated in Figure 7, we visualize the flow paths induced by GCVF and Standard FM. Standard FM exhibits curved evolution paths with pronounced endpoint variation across different inference steps, indicating sensitivity to the number of steps and a reliance on multi-step inference for high-quality outputs. In contrast, our method, which enforces alignment between instantaneous and average velocity fields, yields straight flow paths with consistent endpoints across varying inference steps, enabling high-quality trajectory generation in a single step.

## 5 CONCLUSION

We propose FlashPlanner, a real-time flow-based planner for autonomous driving. FlashPlanner introduces a novel *Globally Consistent Velocity Field* (GCVF) for flow matching, which unleashes the potential of diffusion-based planners. Our GCVF-based training method produces straight flow paths, which enables stable one-step trajectory generation. We additionally give a detailed analysis of the existing design choices of diffusion-based methods and prune inherent redundancy, which further accelerates the diffusion-based planning. FlashPlanner achieves state-of-the-art closed-loop performance on nuPlan and delivers 166 FPS inference speed with only 2.5h training. FlashPlanner offers a more friendly training and inference approach for diffusion-based planners.

## 6 ETHICS STATEMENT

Our research aims to advance the field of autonomous driving and does not present immediate, direct negative social impacts. We believe our work has the potential for a positive impact by improving the safety and reliability of autonomous systems.

The dataset used in this study, namely nuPlan, is publicly available and have been widely adopted by the machine learning community for academic research. All data was handled in accordance with their specified licenses and terms of use. We did not use any personally identifiable or sensitive private information.

We have focused our evaluation on standard academic benchmarks. We encourage future research building upon our work to consider the specific ethical implications of their target applications.

## 7 REPRODUCIBILITY STATEMENT

To ensure the reproducibility of our research, we provide a comprehensive description of our methodology, implementation details, and experimental setup in the main paper. Furthermore, we commit to making our code, pre-trained models, and experiment configurations publicly available upon publication of this paper.

## 8 USE OF LARGE LANGUAGE MODELS (LLMs) STATEMENT

During the preparation of this manuscript, we utilized Large Language Models (LLMs), specifically Claude-4 and GPT-5, as a writing assistance tool. The use of LLMs was limited to improving the grammar, clarity, and readability of the text. This includes tasks such as rephrasing sentences for better flow, correcting spelling and grammatical errors, and ensuring stylistic consistency.

The core scientific ideas, experimental design, results, and conclusions presented in this paper are entirely our own. LLMs were not used to generate any of the primary scientific content or interpretations. The final version of the manuscript was thoroughly reviewed and edited by all authors, who take full responsibility for its content and originality.

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

# A APPENDIX

## A.1 RELATED WORK

**Imitation-based Planner.** Imitation-based AD planners (Bansal et al., 2018; Vitelli et al., 2022; Huang et al., 2023) learn the driving policy by cloning control policies from expert demonstrations. With the abundance and affordability of driving data today, this imitation learning (IL) paradigm achieves remarkable advances due to its scalability and ease of convergence, especially in the end-to-end pipelines (Hu et al., 2023; Chitta et al., 2022; Chen et al., 2024). However, IL methods are vulnerable to issues like covariate shift and mode collapse, which incur performance degradation, especially on the closed-loop setting (Dosovitskiy et al., 2017; Jia et al., 2024). Many techniques (Cheng et al., 2024b;a) have been proposed to address these issues and push the limit of the imitation-based approach. Here we adopt the mid-to-mid fashion, which utilizes post-perception outcomes as input, to focus on the planning task. Despite these efforts, most existing approaches still heavily rely on post-processing to refine or select appropriate trajectories, which means they fail to capture the expert driving policies to some extent.

**Diffusion and Flow Matching.** Diffusion and flow matching have recently gained traction in many research fields because of their multi-modality and high-quality outputs (Song et al., 2021; Lipman et al., 2022; Black et al.). A fundamental dilemma in diffusion models is the tension between inefficient iterative sampling and the need for high-quality outputs (Wang et al., 2025). Lots of alternative training objectives are proposed to mitigate this challenge, such as consistency constraint (Song et al., 2023), average velocity prediction (Geng et al., 2025), and state transition mapping (Wang et al., 2025). Despite these advances, the efficacy of these training objectives remains understudied.

The potential of diffusion and flow matching for planning tasks was first demonstrated in robotics (Black et al.; Intelligence et al., 2025). Several works adapt diffusion models to advance IL-based AD planners but encounter unexpected obstacles such as mode collapse, trajectory divergence, and low efficiency (Liao et al., 2025; Xing et al., 2025; Yang et al., 2024). These obstacles are alleviated through practices from conventional IL methods like introducing prior anchors and auxiliary losses, which result in inefficient training and inference. Some works (Zheng et al., 2025) try to reduce the latency by using DPM-Solver (Lu et al., 2022), a smaller encoder, and jointly modeling ego and other agents, which achieves good performance. Nevertheless, these diffusion-based methods are built atop practices from conventional imitation-based methods, yet the principles for well-designed diffusion/flow-based planners are still under exploration.

In this paper, we introduce the global consistent velocity field, a robust and performant learning objective for diffusion-based planners, enabling stable one-step trajectory generation. We further systematically investigate the components of existing diffusion-based autonomous planners and remove redundant designs. Finally, a lightweight yet performant flow-matching planner is proposed, which delivers strong closed-loop performance with exceptional efficiency.

## A.2 DESIGN CHOICES OF FLASHPLANNER

We summarize the design choices of FlashPlanner, built upon Diffusion Planner, as follows.

- *GCVF (One Step)*: Replace the diffusion training and inference paradigm with GCVF, employing a single inference step.
- *Planning-only*: Focus exclusively on ego planning, removing traffic prediction network modules and their corresponding loss terms from the training objective.
- *Other-As-Ego*: Augment the training dataset with trajectory samples of non-ego vehicles extracted from existing scenarios.
- *MLP Decoder*: Replace the Transformer-based decoder with a compact MLP.
- *Light Encoder*: Reduce the number of self-attention layers in the encoder to one.
- $(x, y, \theta)$: Represent ego future states as $(x, y, \theta)$ instead of $(x, y, \cos\theta, \sin\theta)$.

## A.3 EXPERIMENTAL DETAILS

**Baseline.** We compare FlashPlanner against the following baselines.

- **Log Replay** tracks human trajectories using an LQR controller as the expert baseline.

- **IDM** (Treiber et al., 2000) represents a classic car-following model widely used in traffic simulations.

- **PDM** (Dauner et al., 2023) offers three variants: PDM-Closed generates trajectory proposals using IDM policies with varying hyperparameters and selects the optimal one through rule-based scoring; PDM-Open employs MLPs conditioned on centerlines; PDM-Hybrid combines both approaches.

- **GameFormer** (Huang et al., 2023) models interactive planning and prediction based on level-k game theory, with outputs refined through nonlinear optimization.

- **PLUTO** (Cheng et al., 2024a) extends PDM-Open with contrastive learning for enhanced environmental understanding, followed by rule-based post-processing.

- **Diffusion Planner** (Zheng et al., 2025) applies a diffusion transformer to generate ego trajectories conditioned on vectorized scene representations, with optional refinement.

- **UrbanDriver** (Scheel et al., 2022) utilizes policy gradient optimization with PointNet-based polyline encoders and Transformers.

- **PlanTF** (Cheng et al., 2024b) represents a strong imitation learning baseline built on a Transformer architecture with efficient designs.

A.4    IMPLEMENTATION DETAILS.

FlashPlanner is trained on 4 NVIDIA A100 80GB GPUs with a batch size of 2048 over 100 epochs, including a 5-epoch warmup phase. We employed the AdamW optimizer with a learning rate of 1e-3. To ensure fair comparison and evaluate computational efficiency under practical deployment conditions, all inference experiments were performed on the same device with a single NVIDIA RTX A6000 GPU and an AMD EPYC 7542 32-Core CPU. The detailed hyperparameter configuration is presented in Table 4.

Table 4: Hyperparameters of FlashPlanner

| Parameter | Symbol | Value |
|---|---|---|
| Num. past timestamps | $T_h$ | 21 |
| Num. future timestamps | $T_f$ | 4 |
| Num. neighboring vehicles | $N_d$ | 32 |
| Dim. neighboring vehicles | $D_d$ | 11 |
| Num. static objects | $N_s$ | 5 |
| Dim. static objects | $D_s$ | 10 |
| Num. lanes | $N_l$ | 70 |
| Num. points per polyline | $P_l$ | 20 |
| Dim. lanes | $D_l$ | 12 |
| Num. navigation lanes | $N_r$ | 25 |
| Dim. hidden layer | $D$ | 192 |
| Time sampler | - | Uniform |
| Inference step | $n$ | 1 |

A.5    ADDITIONAL RESULTS

Based on the ablation results in Table 7 and Figure 9, we analyze the contribution of our design choices.

**Ego State:** Prior works represent the ego's future state as $(x, y, \cos\theta, \sin\theta)$, whereas we use $(x, y, \theta)$. The former introduces redundant constraints ($\sin^2\theta + \cos^2\theta = 1$), which are brittle and lead to degraded performance in reactive scenarios.

**Data Preprocessing:** Following Diffusion Planner (Zheng et al., 2025), we apply data augmentation, including ego-centric transformation, z-score normalization, and state perturbation with future

Table 5: Efficiency comparison of learning-based planners.

| Method | Model Size (MB) | FLOPs (G) | Throughput (FPS) |
|---|---|---|---|
| PlanTF | 7.72 | 0.67 | 97 |
| PLUTO w/o refine. | 16.18 | 0.73 | 47 |
| Diffusion Planner | 23.05 | 1.54 | 13 |
| FlashPlanner (Ours) | 4.60 | 0.48 | 166 |

Table 6: **Performance Comparison on nuPlan.** All methods were trained for 100 epochs and evaluated with one-step generation.

| Method | Val14 | | Test14-hard | |
|---|---|---|---|---|
| | NR | R | NR | R |
| MeanFlow | 86.26 | 77.38 | 70.53 | 65.00 |
| Consistency Models | 80.32 | 73.88 | 69.22 | 61.61 |
| GCVF (ours) | 89.19 | 84.6 | 76.46 | 74.12 |

Table 7: **Ablation of each component on nuPlan Test14-hard.** "w/o" denotes "without."

| Component | Setting | NR | R |
|---|---|---|---|
| Baseline | FlashPlanner | 76.46 | 74.12 |
| Ego State | $(x, y, \sin\theta, \cos\theta)$ | 76.96 | 71.32 |
| Data Preprocessing | w/o data augmentation | 68.96 | 61.82 |
| | w/o scenario balancing | 71.54 | 72.52 |
| | w/o other-as-ego | 77.38 | 68.34 |
| Time Sampler | LogNormal(-0.4, 1.0) | 76.06 | 74.06 |
| | Beta(1.0, 1.5) | 75.51 | 70.76 |

trajectory interpolation. Removing data augmentation yields poor performance, highlighting its role in mitigating out-of-distribution shift and enhancing generalization. Additionally, balancing the number of scenarios per type (see Figure 11) in the trainset mitigates dataset skew and consistently improves planning performance.

For the planning-only objective, we augment the 1M training frames by extracting 250K trajectories from neighboring vehicles with significant heading or velocity changes, treating them as pseudo-ego trajectories. This strategy leverages diverse agent behaviors to create more challenging training samples without additional data collection, demonstrating improved generalization capability.

**Time Sampler:** We evaluate three time samplers for flow matching: Uniform $\mathcal{N}(0, 1)$, LogNormal(-0.4, 1.0) that emphasizes intermediate times (Esser et al., 2024), and Beta(1.0, 1.5) that biases toward early times (Black et al.). Results show broad robustness across samplers, with the simplest uniform sampling achieving the best.

**Inference Steps:** To numerically solve ODEs, we use the Euler method, which requires a pre-specified number of steps (Equation (11)). Figure 9 demonstrates that FlashPlanner enables one-step generation of high-quality trajectories for autonomous driving, while supporting refinement with more inference steps.

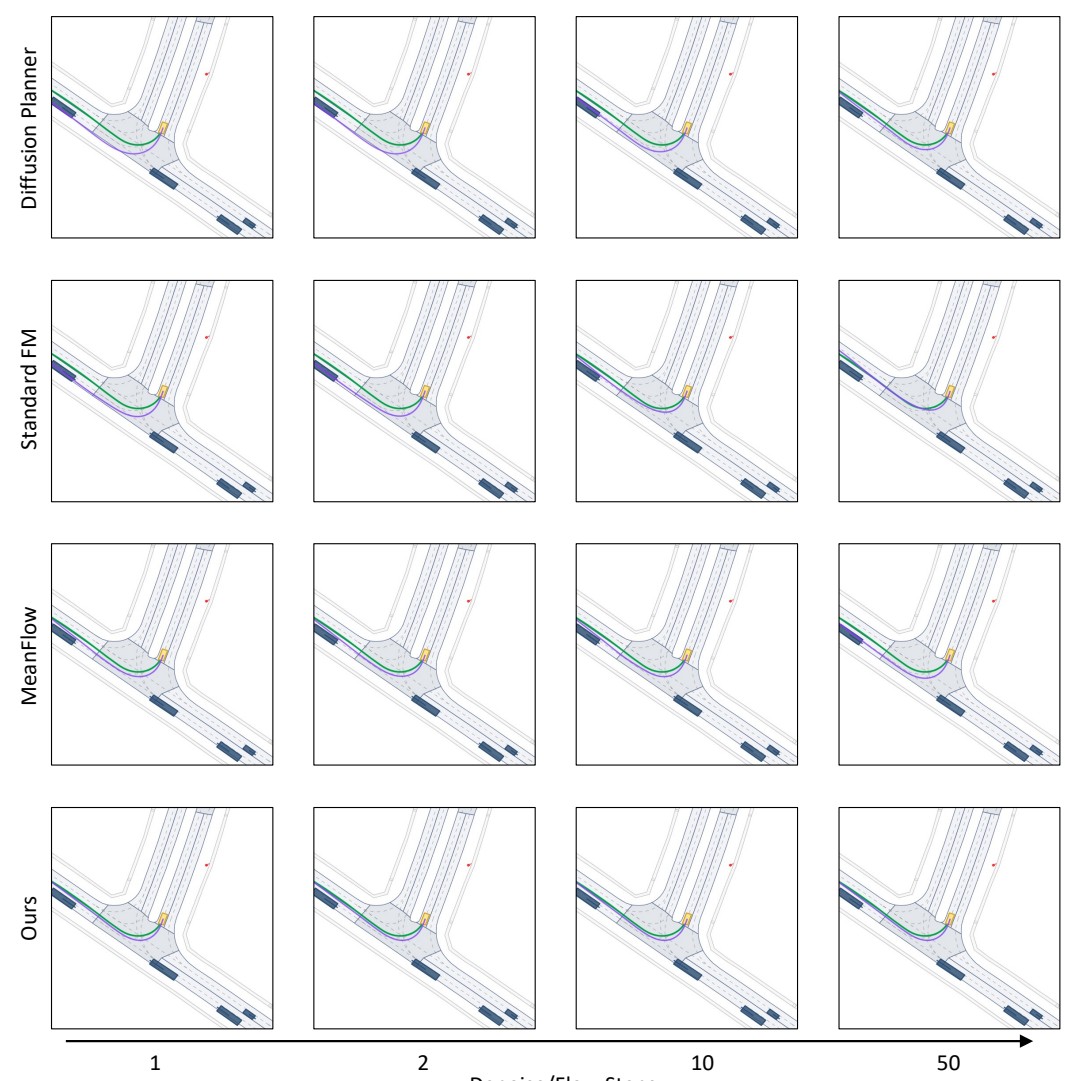

Figure 8: **Future trajectory visualization in closed-loop evaluation.** Trajectory planning results of four methods with varying denoise/flow steps for a challenging narrow-road turning scenario, showing the planned future trajectory and the ground-truth of the ego vehicle. When navigating through dense traffic, our method consistently generates safe and feasible driving trajectories across different inference steps, while other methods result in collisions.

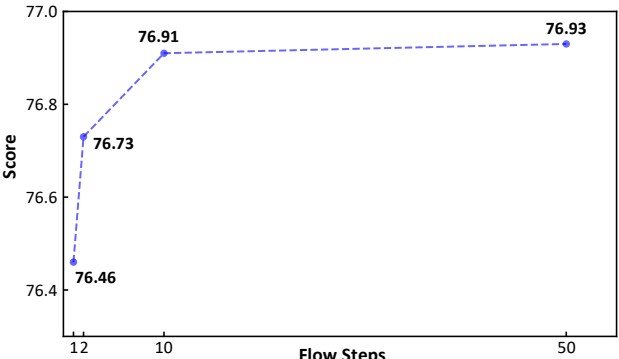

Figure 9: Ablation study on inference steps (Test14-hard NR).

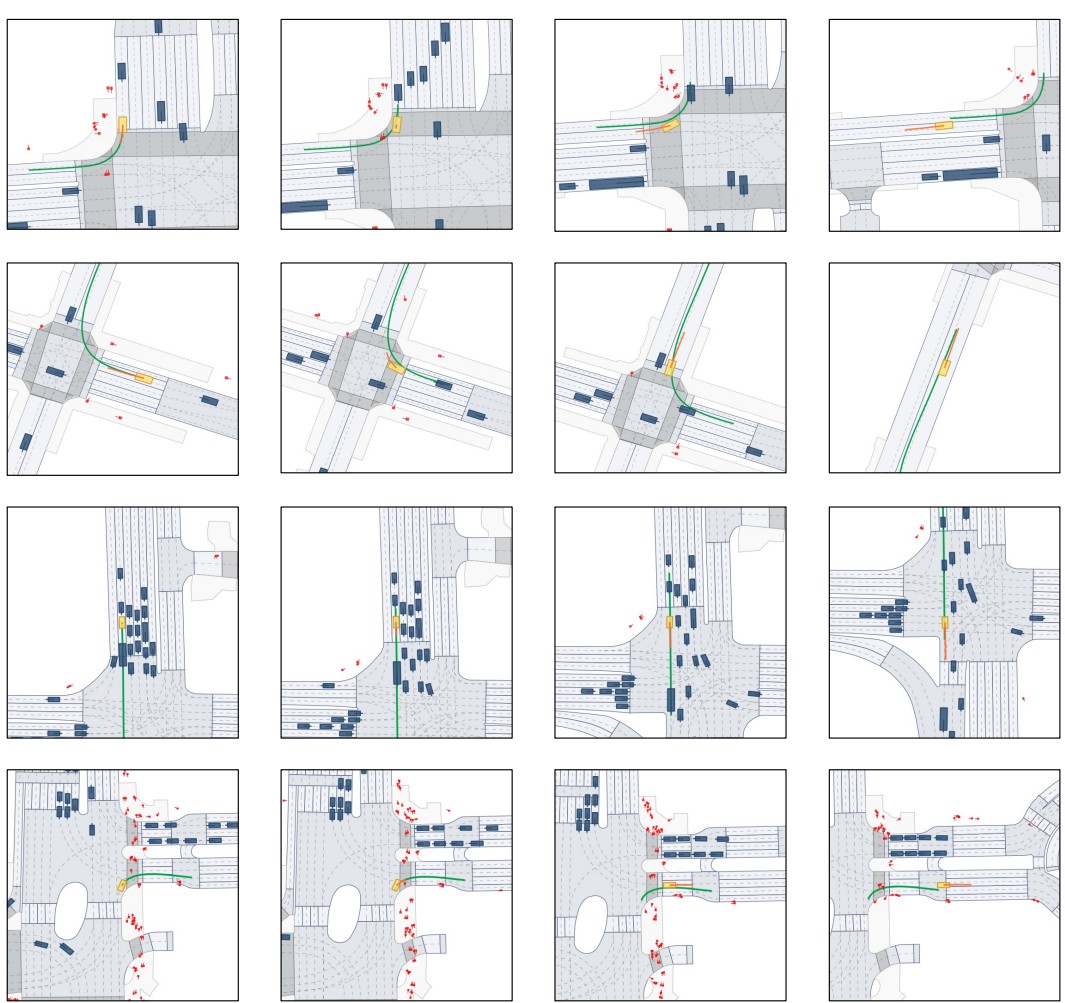

Figure 10: **Visualization of FlashPlanner Closed-loop planning results on hard cases.** Each row represents a scenario at 0, 5, 10, and 15 seconds intervals. Each frame includes the future planning of the ego vehicle and the ground truth ego trajectory.

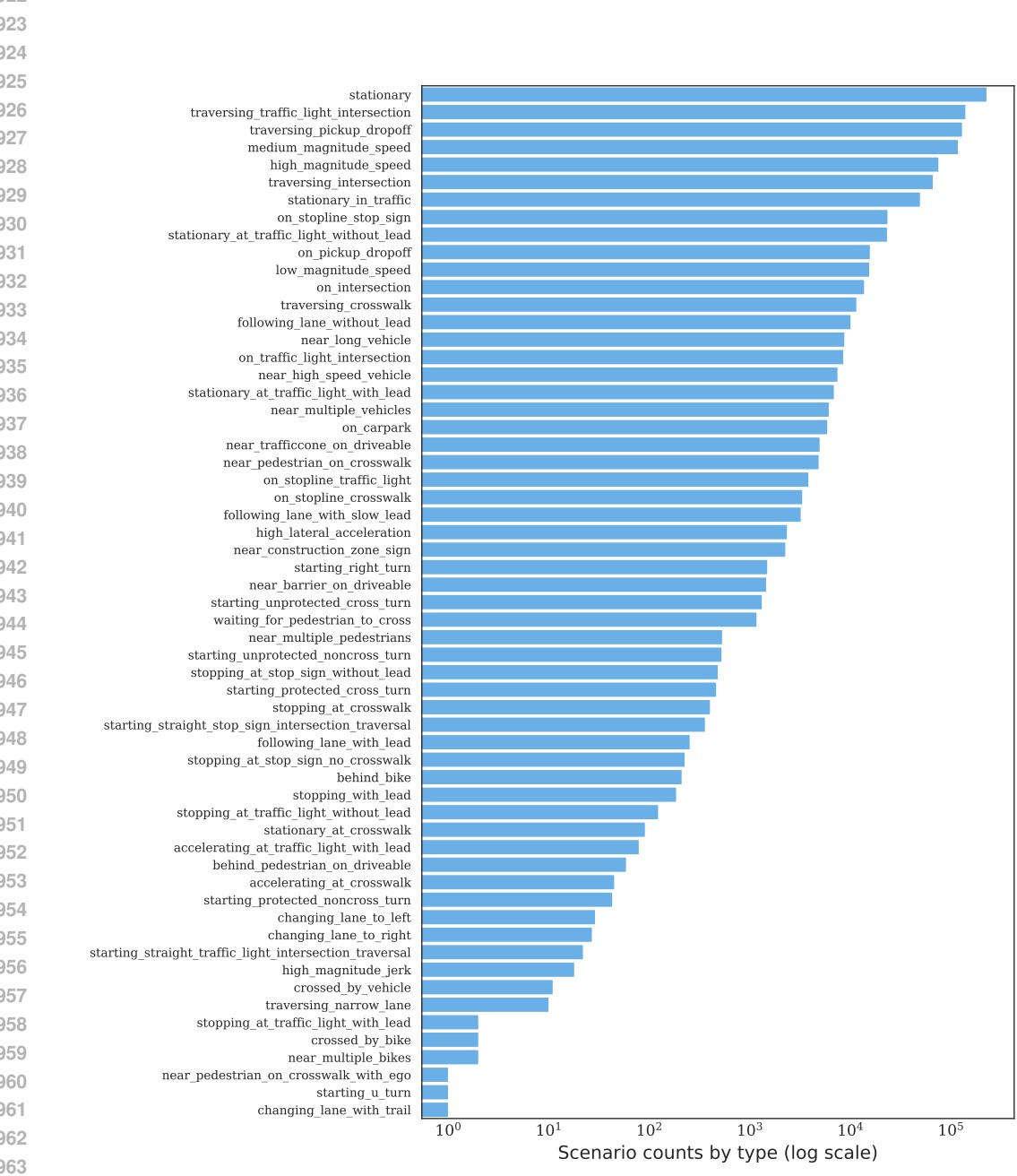

Figure 11: Distribution of scenario types in the training set

