# OpenReview forum: "FlashPlanner: Accelerating Diffusion-based Planner for Autonomous Driving via Globally Consistent Velocity Field and Redundancy Reduction"
_ICLR.cc/2026/Conference — ICLR 2026 Conference Withdrawn Submission_

### Official Review · Reviewer_DNRt · 2025-10-22

**Soundness:** 2
**Presentation:** 3
**Contribution:** 1
**Rating:** 2
**Confidence:** 4

**Summary:**

This paper proposes FlashPlanner, a flow-matching-based planner for autonomous driving that aims to accelerate diffusion-style trajectory generation. The key idea is introducing a Globally Consistent Velocity Field (GCVF), which enforces alignment between instantaneous and average velocities to enable single-step trajectory generation.

It performs a “redundancy reduction” of existing diffusion planners, removing joint modeling of multiple agents, replacing Transformers with MLPs, and simplifying the encoder. Experiments on the nuPlan benchmark show that FlashPlanner achieves 12× faster inference (166 FPS) and state-of-the-art closed-loop performance compared with Diffusion Planner.

**Strengths:**

1. The system achieves real-time closed-loop planning while maintaining competitive accuracy — a clear engineering advance for autonomous driving planners.

2. Experiments cover multiple baselines (rule-based, hybrid, and learning-based) and include closed-loop metrics and qualitative visualizations.

3. The paper provides clear implementation details.

**Weaknesses:**

1. The Globally Consistent Velocity Field (GCVF) objective in this paper essentially reduces to a form of direct endpoint prediction / average-velocity alignment: the model is trained to predict the clean endpoint $x_1$ . The claimed GCVF is not mathematically or conceptually distinct from existing $x_1$ -prediction formulations. Thus, the theoretical contribution is minimal.

2. The overall approach seems to combine multiple practical simplifications (endpoint prediction, lighter architecture, other-as-ego augmentation) without providing insight into their interactions or theoretical rationale. While this yields efficiency gains, such lightweight redesigns are already common in recent planning and diffusion-based architectures. As presented, the architectural simplification reads as a collection of engineering heuristics rather than a principled design, and the absence of systematic ablation or interpretability analysis weakens the claimed insights.

**Questions:**

1. What is the essential difference between GCVF and direct trajectory prediction without flow matching?

1. Could you provide detailed analysis and experiments on the Other-As-Ego strategy, e.g., its effect on interaction-heavy scenarios, whether it truly preserves inter-agent dynamics and what can model learn from other agent?

2. The lightweight encoder causes noticeable performance drops. Could you analyze how it affects scene representation, and why applying the following changes alleviates this issue?

---

### Official Review · Reviewer_7gtJ · 2025-10-28

**Soundness:** 1
**Presentation:** 3
**Contribution:** 2
**Rating:** 2
**Confidence:** 5

**Summary:**

This paper introduces FlashPlanner, an approach that empirically investigates the training of diffusion planners. By employing a flow matching loss and several architectural modifications, it achieves state-of-the-art performance on the nuPlan benchmark.

**Strengths:**

The paper is well-written and easy to follow, complemented by clear and effective visualizations.

**Weaknesses:**

1. The proposed GCVF is equivalent to the data prediction loss used in flow matching [1] and has been applied in existing literature [2]. Therefore, the authors' claim of novelty in proposing this method and their extensive discussion of the mean flow may constitute an overstatement.

2. While this paper demonstrates that certain modifications to the diffusion planner can improve performance on the nuPlan benchmark, these designs lack a thorough rationale analysis. Their effectiveness might be specific to this benchmark, acting as a "hack". For instance, the use of a single-layer self-attention ("light encoder") in FlashPlanner could be a benchmark-specific tuning rather than a generally optimal design. Thus, the generalizability of these modifications requires further validation on more diverse benchmarks.

[1] Lipman Y, Havasi M, Holderrieth P, et al. Flow matching guide and code[J]. arXiv preprint arXiv:2412.06264, 2024.

[2] Tan T, Zheng Y, Liang R, et al. Flow Matching-Based Autonomous Driving Planning with Advanced Interactive Behavior Modeling[J]. arXiv preprint arXiv:2510.11083, 2025.

**Questions:**

N/A

---

### Official Review · Reviewer_7kcA · 2025-10-31

**Soundness:** 3
**Presentation:** 3
**Contribution:** 3
**Rating:** 8
**Confidence:** 3

**Summary:**

This paper introduces globally consistent velocity field as the training objective for flow matching, unleashing the potential of diffusion-based planners, and by removing redundant design choices, the framework enables fast and high-quality trajectory generation in closed-loop planning.

**Strengths:**

* A new training objective Globally Consistent Velocity Field (GCVF) is proposed for flow-matching, enabling both one-step fast trajectory generation and refinement with more inference steps.
* With the GCVF unleashing the potential of diffusion-based planner, the paper provides a systematically analysis about design choices and prune redundant ones, leading to a lightweight planner with fast inference speed.

**Weaknesses:**

* The main results miss the metrics on Test14, making the evaluation not complete.

**Questions:**

* In Fig.3, what is the difference between last two variants?
* Can GCVF be applied to other fields, e.g. image generation?

---

### Note · Authors · 2025-11-14

I have read and agree with the venue's withdrawal policy on behalf of myself and my co-authors.